# Comprehensive profiling of the ligand binding landscapes of duplexed aptamer families reveals widespread induced fit

Jeffrey D. Munzar[1,2], Andy Ng[1,2] & David Juncker [1,2,3]

Duplexed aptamers (DAs) are ligand-responsive constructs engineered by hybridizing an aptamer with an aptamer-complementary element (ACE, e.g., a DNA oligonucleotide). Although DAs are commonly deployed, the binding dynamics of ternary ACE-aptamer–ligand systems remain underexplored, having been conventionally described by a conformational selection framework. Here we introduce aptamer-complementary element scanning (ACE-Scan) as a method to generate comprehensive hybridization, spontaneous off-rate, and induced fit ligand-binding landscapes for entire DA families. ACE-Scan reveals induced fit in DAs engineered from small molecule- and protein-binding DNA and RNA aptamers, as well as DAs engineered from the natural *add* riboswitch aptamer. To validate ACE-Scan, we engineer solution-phase ATP-specific DAs from 5 ACEs with varying spontaneous and induced fit off-rates, generating aptasensors with 8-fold differences in dynamic range consistent with ACE-Scan. This work demonstrates that ACE-Scan can readily map induced fit in DAs, empowering aptamers in biosensing, synthetic biology, and DNA nanomachines.

[1] McGill University and Genome Quebec Innovation Centre, McGill University, Montreal, H3A 0G1 Quebec, Canada. [2] Department of Biomedical Engineering, McGill University, Montreal, H3A 2B4 Quebec, Canada. [3] Department of Neurology and Neurosurgery, McGill University, Montreal, H3A 2B4 Quebec, Canada. Correspondence and requests for materials should be addressed to D.J. (email: david.juncker@mcgill.ca)

Nucleic acid aptamers, typically obtained through SELEX[1,2], are affinity binders used in a diverse range of applications in biosensing and synthetic biology. A common aptamer-based biosensor format is the duplexed aptamer (DA), a ligand-responsive construct that is engineered by hybridizing an aptamer sequence with an aptamer-complementary element (ACE), such as a short DNA oligonucleotide, to form a synthetic switch[3]. In a DA, the aptamer acts as the ligand binder, while the ACE, which initially hybridizes to a defined portion of the aptamer, acts as a competitive binder and generates a signal upon ligand-dependent dehybridization. This duplex-based design is also shared by naturally occurring riboswitches, which contain a conserved small-molecule-sensing aptamer domain that is coupled to a downstream modular RNA expression domain via hybridization, with ligand-mediated duplex disruption effecting transcriptional or translational control[4,5].

Within the context of folding funnels and free energy basins[6], the dynamics of ligand-binder systems can be broadly described by conformational selection or induced fit. As applied to DA-based systems (e.g., for surface-based DAs, see Fig. 1a), conformational selection assumes that the ACE first dehybridizes from the DA to yield the free aptamer, which then binds the ligand. In contrast, under induced fit, a DA is modeled as actively sensing the ligand from the duplexed state, with ligand-binding catalytically disrupting the duplex and yielding an aptamer bound to its ligand. This mechanistic framework of conformational selection and induced fit pathways for DAs thus mirrors first order ($S_N1$) and second order ($S_N2$) nucleophilic substitution reaction mechanisms, respectively.

Conventionally, DA-based systems have been widely modeled on the basis of conformational selection, with affinities based on the hybridization free energy of the ACE-aptamer duplex (as described for e.g. aptasensor-[7–10], shRNA-[11], and ribozyme-based[12] synthetic DAs, and natural riboswitches[13]). Interestingly, this prevailing model of DA-ligand-binding stands in contrast with the modeling of native aptamers, many of which are known to bind their ligand via induced fit[14]. Furthermore, we recently discovered that some ACEs give rise to, and regulate, induced fit ligand-binding in a small set of DAs engineered from an ATP DNA aptamer, with binding affinities one million-fold higher than predicted by conformational selection[15]. However, DA-to-ligand binding remains underexplored, and it is not yet understood whether ACE-regulated induced fit is an exception specific to the ATP DNA aptamer and subset of ACEs tested[15], or if induced fit also arises in (i) ATP DNA DAs using other ACEs, or in (ii) DA families engineered from other DNA and RNA aptamers.

A major challenge to studying DAs lies in their highly combinatorial design space, as thousands of ACEs, varying in length, location, and/or incorporating mismatched bases, could be used to engineer a ligand-specific DA. Currently, DA constructs are largely designed by trial and error, and studied individually, most often by testing a handful of candidate ACEs for an aptamer of interest. Indeed, there are no methods capable of rapidly and comprehensively screening the binding dynamics of DAs engineered from the compendium of candidate ACEs for a particular aptamer, as existing techniques such as NMR, nuclease resistance assays, single-molecule FRET, and optical tweezers can only measure one construct at a time. Microarrays and NGS flow-cell-based methods are powerful tools to study the equilibrium binding between oligonucleotide probes and a functional nucleic acid of interest[16,17], to study the affinity and sequence space of aptamers[18–20], and to systematically study the affinity landscapes and binding kinetics of binary DNA-protein[21,22] and RNA–protein[23,24] systems. However, ligand binding in DAs

constitutes a ternary ACE-aptamer–ligand system that cannot be studied using existing high-throughput methods.

Here we introduce aptamer-complementary element scanning (ACE-Scan) as a method to systematically and comprehensively profile DA ligand-binding landscapes. ACE-Scan leverages DNA microarray technologies, together with a non-equilibrium assay, to study the binding kinetics of thousands of surface-assembled DAs at once. We apply ACE-Scan to map the binding landscapes of DAs engineered from three DNA and RNA aptamers selected for in vitro against small-molecule ligands, as well as an aptamer against the full-length human protein alpha thrombin. We also profile ACE-duplexed constructs of a natural aptamer, namely the translation regulating *add* riboswitch aptamer from the pathogenic bacteria *Vibrio vulnificus*. Unexpectedly, ACE-Scan of these aptamers reveals unique induced fit ligand-binding profiles in four out of five DA families, together with rich aptamer-specific ACE hybridization and ACE spontaneous off-rate landscapes. To further validate ACE-Scan, we engineer five solution-phase ATP DNA DAs from ACEs with a range of spontaneous and induced fit off-rates. The response of the 5 solution-phase DAs to ligand is found to be consistent with expectations based on ACE-Scan profiles, with the best DA exhibiting a dynamic range nearly one order of magnitude higher than a published DA. These findings improve our fundamental understanding of DA-based systems, with important implications for the design of biosensors, synthetic biology circuits and DNA nanomachines.

## Results

**ACE-Scan microarray design and workflow.** We introduce ACE-Scan, which differentiates between conformational selection and induced fit ligand-binding pathways in DAs using a non-equilibrium surface-based assay[15] (Fig. 1 and Methods section). ACEs ranging from 7 bases up to 32 bases long (in this study) were synthesized on a DNA microarray so as to tile an entire aptamer sequence, allowing for thousands of ACE-aptamer combinations to be queried simultaneously and the binding landscapes of entire DA families mapped in a single experiment (Fig. 1b). Owing to the high dilution of dissociated aptamer molecules, duplex dissociation during ACE-Scan was essentially irreversible, and thus decreases in fluorescence signal constitute an accurate measure of DA dissociation rates. Specifically, for each DA on the microarray, we simultaneously measured duplex dissociation rates under buffer-only incubation ($k_{off}$, which provides a baseline within the conformational selection pathway), and under ligand conditions ($k^{*}_{off,[Ligand]}$), in which increased dissociation over buffer-only $k_{off}$ is ascribed, by definition, to the induced fit pathway[15] (Fig. 1c). By implementing ACE-Scan with different ligand concentrations, the induced fit ligand-binding affinity ($K_{Fit}$) of each DA on the microarray can also be determined[15]. Here we used 42k feature DNA microarrays with 6 identical 7000-spot sub-arrays (Fig. 1d and Methods section), allowing us to profile ligand binding in DA families engineered with 1000–1400 ACEs, each of which was synthesized with 5–7 replicates per sub-array.

To carry out an ACE-Scan experiment, first 0.5 µM of Cy3-labeled aptamer in hybridization buffer was hybridized to all microarray sub-arrays overnight (Methods sextion). After brief washing, green channel fluorescence was recorded to obtain DA-specific hybridization signals ($F_{Hyb}$), which are a direct measure of the hybridization affinity of each ACE ($K_{Hyb}$). Next, each sub-array was incubated for 1 h with either (i) buffer only (BufferOnly sub-array), or (ii) buffer supplemented with a defined ligand concentration ( + [Ligand] sub-array), or (iii) left unincubated (Calibration sub-array). Following a second brief washing, microarrays were imaged in the green channel to quantify the

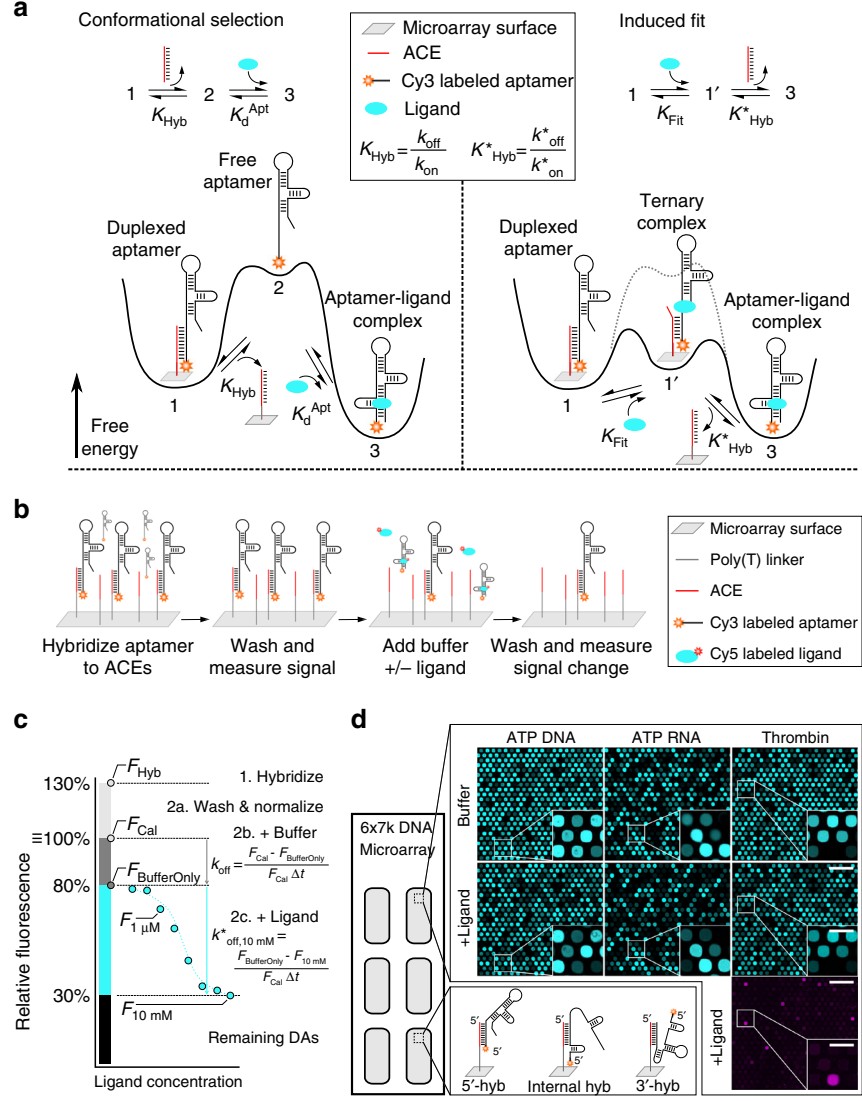

**Fig. 1** ACE-Scan comprehensively profiles the ligand-binding landscapes of duplexed aptamers. **a** Free energy landscape of a surface DA following conformationally selective (left) or induced fit ligand binding (right, conformational selection pathway shown as dotted gray line). Induced fit binding relies on ligand recognition by the ACE-duplexed aptamer, further implying that the hybridized ACE may actively participate in the ternary DA-ligand-binding interaction. **b** ACE-Scan workflow. 1000's of ACEs synthesized on a DNA microarray were first hybridized with the aptamer, and the fluorescence intensity ($F_{Hyb}$) was imaged to assess the hybridization affinity ($K_{Hyb}$) of each ACE. Sub-arrays were either left unincubated (Calibration), incubated with buffer (BufferOnly), or incubated with varying ligand concentrations ( + [Ligand]), and changes in fluorescence were recorded. **c** Schematic of relative fluorescence signals obtained from ACE-Scan. $F_{Cal}$ was defined as 100% for each DA (see Methods section), and changes in relative fluorescence after 1 h ($\Delta t$) were used to calculate $k_{off}$ (spontaneous duplex dissociation rate, units of $h^{-1}$), $k^*_{off}$ (induced fit duplex dissociation rate, units of $h^{-1}$) and $K_{Fit}$ (induced fit binding affinity, units of M). Representative ACE-Scan relative fluorescence values for $F_{Hyb}$ and $F_{BufferOnly}$ are plotted, and a representative dilution curve for 8 ligand concentrations is shown in cyan, with two fluorescence signals labeled ($F_{1 \mu M}$, $F_{10 mM}$). Importantly, DA dissociation rates in ACE-Scan are a sum of $k_{off}$ and $k^*_{off[Ligand]}$ rates, and $k^*_{off[Ligand]}$ is therefore calculated based on the increased dissociation rate, relative to $k_{off}$, induced by the presence of ligand[15]. **d** Representative ACE-Scan images for three in vitro-selected aptamers (ATP DNA, ATP RNA, and thrombin DNA aptamers). Top row of panels: Cy3 fluorescence measured after buffer-only incubation for 1 h ($F_{BufferOnly}$). Middle row of panels: Cy3 fluorescence measured after 10 mM ATP ($F_{10 mM}$) or 2 $\mu$M thrombin ($F_{2 \mu M}$) incubation for 1 h. Bottom right panel: red fluorescence of AF647-labeled thrombin bound to thrombin DAs after incubation with 2 $\mu$M thrombin for 1 h. Scale bars: 300 $\mu$m (large panels) and 100 $\mu$m (insets). Bottom left panel: schematic of DAs engineered with 5′-, 3′- and internally-hybridizing ACEs

remaining number of Cy3-labeled aptamer molecules, yielding $F_{Cal}$, $F_{BufferOnly}$, and $F_{[Ligand]}$, which are used to calculate $k_{off}$, $k^*_{off}$ and $K_{Fit}$ (Fig. 1c and Methods section). When using an AF647-labeled ligand, microarrays were also imaged in the red channel to quantify the number of ligand molecules bound by DAs (Fig. 1d). Finally, the data sets were analyzed using a calibration procedure that corrects for inter-sub-array differences in experimental conditions (Methods section).

**ACE-Scan of the classical ATP DNA aptamer.** We first studied the binding landscape of DAs engineered from the Huizenga and Szostak ATP DNA aptamer[25], as this aptamer has been extensively studied, is widely used for proof-of-concept biosensor designs, and was the first to be engineered as a DA[3]. Furthermore, we recently discovered a small set of 5′-duplexing ACEs regulating induced fit in a handful of ATP DNA DAs[15]. The native aptamer is 27 bases long, and cooperatively binds two ATP

molecules via the formation of a two-site shared binding pocket[26]; a 5-base 5′ extension[3] was added for this study (Fig. 2a).

As described above, we first measured the hybridization affinity of ACEs with differing lengths of complementarity to the ATP DNA aptamer (7–32 bases). 5′-hybridization heat maps show that ACEs with increasing length were more stable, as were those hybridizing towards the 5′ extension, which serves as a toehold for duplex formation (Fig. 2b). Overall, DA hybridization signals

correlated with predicted ACE-aptamer duplex hybridization free energies (Methods section), with a slight dependence on ACE self-complementarity (Supplementary Fig. 1), suggesting that the secondary structure of the ligand-free aptamer minimally impacts ACE hybridization, consistent with our past study[15].

Buffer-only incubation (Methods section) revealed a $k_{off}$ landscape in which ACEs hybridized towards the 5′ end of the aptamer, as well as a collection of 10- to 14-mer ACEs hybridized

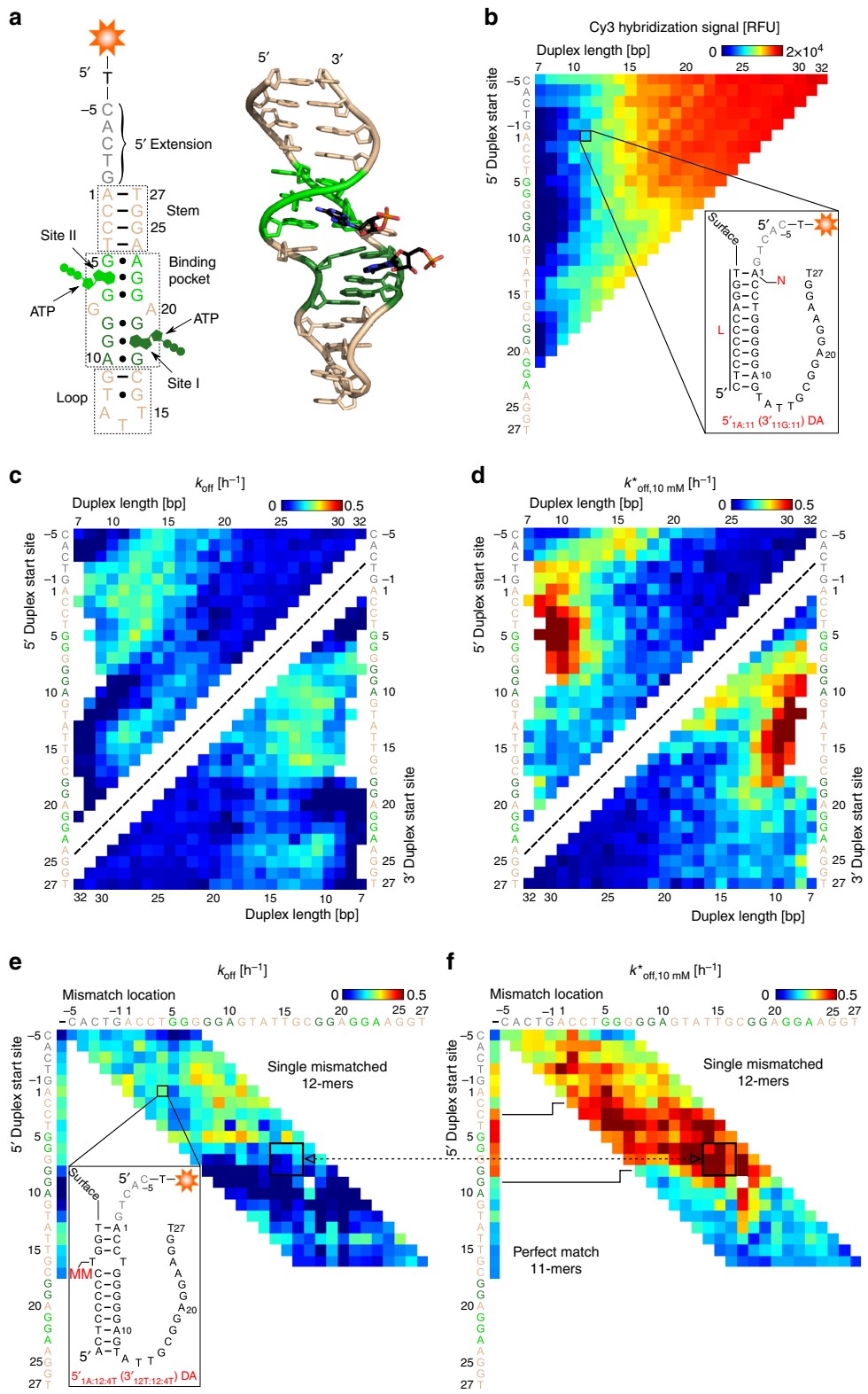

at the 3′-end, exhibited moderate dissociation rates (Fig. 2c shows a combined 5′ and 3′ enantio heat map of $k_{off}$). These regions of the aptamer form a common stem in solution, which likely competes with ACE hybridization and thus increases the dissociation rate of these DAs. In contrast, ACEs with common 3′ start sites within the loop and binding site I regions, and short ACEs hybridized to the 5′ extreme of the 5′ extension, showed low $k_{off}$ rates under buffer-only conditions, as expected.

In parallel, ATP DNA DAs were incubated with 10 mM ATP, which revealed induced fit ligand binding as an additional loss in fluorescence compared to buffer only ($k^{*}_{off,10\,mM}$ landscape, Fig. 2d). 7- to 12-mer ACEs hybridizing to site II formed DAs with a high propensity for induced fit. Furthermore, for 10- to 15-mer ACEs, a trend of ACEs promoting induced fit with common 3′ start-sites, hybridizing towards the 5′ extension of the aptamer, was observed (5′$_{1A:10}$ to 5′$_{-5C:15}$, or 3′$_{10A:10}$ to 3′$_{10A:15}$). These ACEs hybridize to guanosine bases present in binding site II of the aptamer, supporting our previous observation that ACE hybridization of site II promotes induced fit in ATP DNA DAs[15]. Corroborating this model, 8- to 11-mer ACEs hybridized towards the 5′ extension of the aptamer, leaving site II increasingly free, displayed much lower induced fit binding. We also investigated the effect of ACE spacer length and ACE surface density on ATP DNA DA binding profiles, as well as control experiments using GTP as a non-specific ligand, and found them to be consistent with expectations (Supplementary Note 1 and Supplementary Figs. 2–4), thus validating the above findings. While outside the scope of this work, it would be interesting to consider the order of binding of ATP molecules to ATP DNA DAs, which could have implications for engineering DAs with controlled cooperativities, as demonstrated for the native ATP DNA aptamer[27].

The $k^{*}_{off,10\,mM}$ landscape shown in Fig. 2d differs significantly from the hybridization (Fig. 2b) and $k_{off}$ (Fig. 2c) landscapes. Furthermore, DA induced fit binding is correlated neither to the DNA hybridization free energy of ACE-aptamer duplexes, nor to the predicted self-hybridization free energies of ACEs (Supplementary Fig. 5). Hence, induced fit does not appear to be predictable based on current knowledge, implying that experiments, such as those shown here, are required to map the induced fit binding landscape of DAs. Additionally, ACE-Scan can be used to identify ligand-responsive surface DA biosensors with high signal-to-noise ratio. For example, 3′$_{18G:10}$ to 3′$_{18G:12}$ (5′$_{9G:10}$ to 5′$_{7G:12}$) ACEs form stable DAs with low $k_{off}$, and yet these DAs also exhibit a high dissociation rate under 10 mM ATP (Fig. 2d), making these ACEs good candidates for engineering surface biosensors.

To further validate ACE-Scan as a powerful method for DA screening, we evaluated whether ACE-Scan could quantitatively measure the induced fit ligand-binding affinity ($K_{Fit}$) and maximum induced fit dissociation rate ($k^{*}_{off,max}$) of DAs. We generated a dilution curve by combining ligand-response data obtained on two microarray slides (total of 12 sub-arrays and 8 ATP dilutions, ranging from 10 mM to 0.124 μM), and extracted comprehensive $K_{Fit}$ and $k^{*}_{off,max}$ landscapes for ATP DNA DAs by fitting a two-parameter nonlinear regression to the dilution curve (Methods section and Supplementary Fig. 6). We observed similar induced fit ligand-binding affinities of 9- to 12-mer DAs for ATP as previously measured using an in-house DA surface-based assay ($K_{Fit}$ of ~200 μM for ATP-responsive DAs[15], Supplementary Fig. 6). As expected, the $k^{*}_{off,max}$ landscape obtained from the dilution curve fit (Supplementary Fig. 6) was similar to the above-reported $k^{*}_{off,10\,mM}$ landscape (Fig. 2d), implying that a single high concentration of ligand can be used to accurately profile DA families for induced fit binding.

Taking advantage of the high capacity of the DNA microarray, we also studied all possible 12-mer (Fig. 2e, f and Supplementary Fig. 7) and 15-mer (Supplementary Fig. 7) ACEs with single base-pair mismatches to the aptamer at every position. Mismatches are expected to destabilize the hybridized ACE and potentially restructure the 3D ACE-aptamer duplex, both of which could introduce additional propensity for induced fit ligand binding. Generally, single mismatched 12-mer ACEs displayed a similar, but slightly higher, $k_{off}$ landscape to perfect match 12-mer ACEs (Fig. 2e), whereas single mismatched 12-mer ACEs exhibited increased induced fit ligand binding as compared to the same perfect-match 12-mer or 11-mer ACE (Fig. 2f). Interestingly, it was possible to identify subsets of single-mismatched ACEs with low $k_{off}$ and increased $k^{*}_{off,10\,mM}$ as compared to the performance of perfect match 11-mers (e.g., dashed line between 5′$_{6G-8G:12:14T-17G}$ DAs, Fig. 2e, f, and see Supplementary Fig. 7). These results underline ACE mismatches as a factor regulating DA induced fit binding, and supports ACE-Scan experiments with mismatched ACEs as a powerful method to identify DAs with induced fit binding properties superior to those of perfectly matched DAs, such as higher sensitivity and signal-to-noise ratio.

**Validating ACE-Scan for ATP DNA DA solution-phase bio-sensors.** Each of the 1,542 microarray-based ATP DNA DA constructs tested in this study represents a quantitative ligand-responsive fluorescence surface biosensor that can be used directly for ATP readout based on our non-equilibrium surface assay[15] (for performance of select 9-mer surface DAs, see Supplementary Fig. 6). To further validate that ACE-Scan $k_{off}$ and $k^{*}_{off}$ landscapes are reflected in the performance of DAs designed for other assay formats, we conducted solution-phase assays using DAs engineered to signal based on Förster resonance energy transfer (FRET) (Fig. 3a). The solution FRET DA format was chosen for validation as it is commonly employed in the literature, straightforward to implement, compatible with a range of

**Fig. 2** ACE-Scan generates rich ligand-binding landscapes for the ATP DNA DA family. **a** Left, secondary structure of the ATP DNA aptamer, with functional regions labeled, shared binding pocket and ligand highlighted in dark (site I) and light (site II) green, and 5′ extension in gray. Right, 3D view of the ATP DNA aptamer consensus sequence bound to two ATP molecules (PDB 1AW4[26]). **b** Hybridization signal heat map of ATP DNA DAs for all 7- to 32-mer ACEs, obtained after 16 h of hybridization. In this 5′ heat map representation, each tile represents an experimental surface-based DA biosensor in the form 5′$_{N:L}$, representing an ACE of length L (columns) that hybridizes starting at 5′ base N within the aptamer sequence (rows). Tiles are positioned at the 5′ location of ACE hybridization within the aptamer sequence, with tiles lying on southwest by northeast diagonals representing DAs with common 3′ duplex start sites and increasing ACE length (in the format 3′$_{N:L}$). The inset shows the 5′$_{1A:11}$ (or equivalently, 3′$_{11G:11}$) DA. Missing white tiles represent DAs that did not pass quality control, and which were subsequently removed from the data set (Methods section). **c, d** Enantio heat maps, i.e., combined 5′ and 3′ heat map representation, of the response of ATP DNA DAs to incubation with (**c**) buffer only or (**d**) 10 mM ATP. **e, f** ACE-Scan of single mismatched 12-mer ACEs under (**e**) buffer-only or (**f**) 10 mM ATP conditions. Each tile represents a DA with an ACE of length L that forms a duplex starting at 5′ base N (rows) and that is mismatched at base MM (columns) within the aptamer sequence, in the form 5′$_{NL:MM}$. The leftmost column represents perfect match 11-mer ACEs, which are structurally similar to 5′-terminally mismatched 12-mer ACEs (solid lines in **f**). The dotted line highlights the relatively high ligand sensitivity of 5′$_{6G-8G:12:14T-17G}$ DAs (black boxes in **e**, **f**)

instruments for readout, and versatile in the sense that solution FRET DAs can be readily converted to signal via other formats (e.g., fluorescence, nucleic acid circuits, or colorimetry). Importantly, however, whereas ACE-Scan decouples conformational selection from induced fit for DAs implemented on a surface (with ligand-specific signaling a function of only induced fit), the response of homogeneous solution-phase FRET DAs to ligand is expected to be a function of both conformational selection (i.e., $k_{off}$) and induced fit (i.e., $k^*_{off}$) ligand-binding pathways[15], as

these pathways co-exist within a four-state thermodynamic cycle in solution (Fig. 3a). As such, the most ligand-sensitive solution FRET DAs are expected to be those with both high $k_{off}$ (i.e., shorter ACEs with lower hybridization free energies) and high $k^*_{off}$ (i.e., ACEs that promote induced fit).

For validation, we selected four ACEs that differed in $k_{off}$ and $k_{off,10\,mM}$ values, engineered solution-phase FRET DAs, and compared the performance of the four DAs to a published 15-mer ATP DNA DA[28] identified by ACE-Scan to have moderate $k_{off}$ and low $k^*_{off,10\,mM}$ values. Solution FRET DAs (labeled in the form 5′Q$_{N:L:MM}$) were engineered using the same 5′-Cy3-labeled ATP DNA aptamer construct used for ACE-Scan, and ACEs were labeled with a 3′ Black Hole Quencher (BHQ-2) compatible with Cy3 FRET measurements. The sequences of the tested ACEs are shown overlaid the $k_{off}$ and $k^*_{off,10\,mM}$ landscapes and ATP DNA aptamer sequence (Fig. 3b, c), and consist of (1) the 5′Q$_{-2T:9}$ ACE, with moderate $k_{off}$, moderate $k^*_{off,10\,mM}$, and documented induced fit binding[15], (2) the 5′Q$_{4T:10}$ and (3) 5′Q$_{4T:12:6G}$ ACEs, both with low $k_{off}$ and high $k^*_{off,10\,mM}$, (4) the previously published 5′Q$_{1A:15}$ ACE[28], and (5) the 5′Q$_{-5C:9}$ ACE, which was selected as a negative control as it exhibits low $k_{off}$, low $k^*_{off,10\,mM}$, and inhibited induced fit[15].

The solution FRET DAs displayed a more than 8-fold range of fluorescence responses under 10 mM ATP (Fig. 3d). Overall, the relative performance of the five DAs was in qualitative agreement with the ACE-Scan data sets; as expected, the negative control 5′Q$_{-5C:9}$ DA exhibited the weakest ligand-specific response, whereas the short 5′Q$_{-2T:9}$ DA, which exhibits induced fit and which has a high spontaneous off rate, displayed the highest dynamic range. The 10-mer and single-mismatched 12-mer DAs, both of which exhibit low $k_{off}$ and high $k^*_{off}$, exhibited good performance, as expected, whereas the previously published 15-mer DA (5′Q$_{1A:15}$) displayed poor performance, similar to that of the negative control 9-mer DA, and in-line with the moderate $k_{off}$ and low $k^*_{off,10\,mM}$ values measured by ACE-Scan for the 15-mer DA. On the basis of the $k_{off}$ landscape, the 5′Q$_{-2T:9}$ DA is expected to have a correspondingly higher population of free aptamer molecules, which may explain why the 5′Q$_{-2T:9}$ DA generates an increased FRET signal over the 5′Q$_{4T:10}$ or 5′Q$_{4T:12:6G}$ DAs when deployed in solution.

Interestingly, whereas shorter ACEs have conventionally been used to engineer DAs with increased sensitivities on the basis of a conformationally selective ligand-binding model[8,15], here the duplex location of the ACE is shown to significantly impact the response of solution-phase DA biosensors, in line with the induced fit ACE-Scan landscape. Most striking is the difference in dynamic range of the two 9-mer constructs tested here, as these

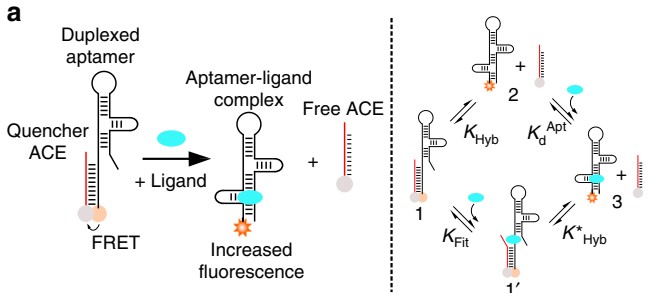

**a**

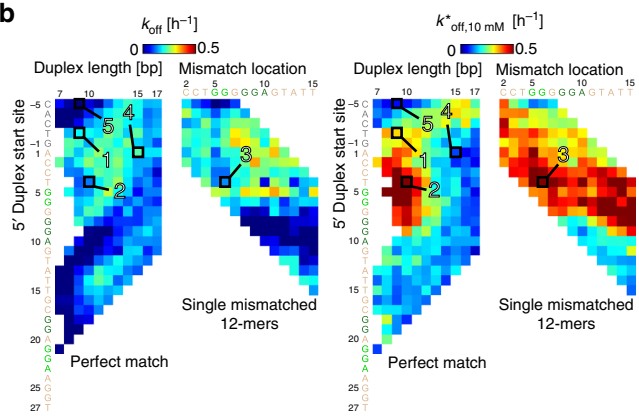

**b**

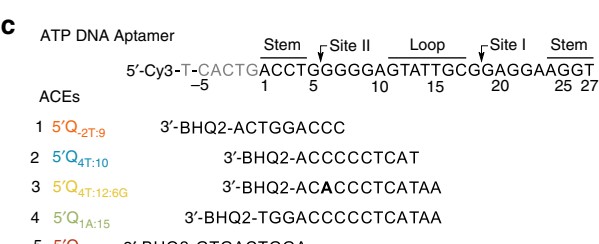

**c**

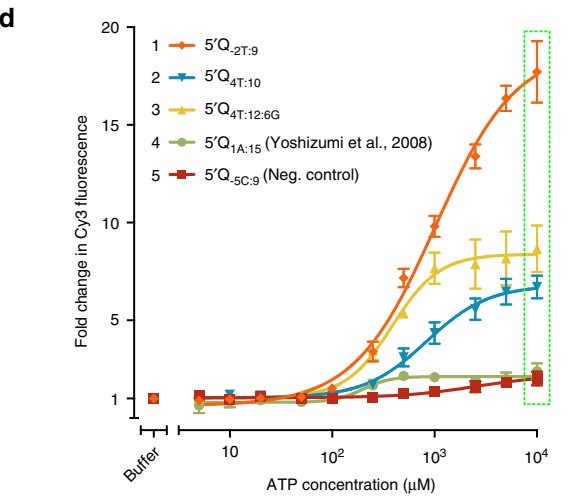

**d**

**Fig. 3** Validation of ACE-scan with a FRET-based solution-phase assay. Five ATP DNA DAs were re-engineered as FRET DAs and solution-based binding curves were compared to ACE-scan heat map results. **a** Overview of the solution FRET assay. A four-state, two-pathway thermodynamic cycle (-**1**-**2**-**3**-**1**'-) is shown schematically for solution FRET DAs. **b** The five tested ACEs are shown schematically as boxed cells on the $k_{off}$ and $k^*_{off,10\,mM}$ ACE-Scan landscapes (heat maps are truncated from Fig. 2). **c** Sequences of the five ACE constructs tested. The 5′Q$_{1A:15}$ ACE is adapted from the literature[28] and is representative of the wide variety of published ATP DNA DAs. The mismatched adenosine in the 5′Q$_{4T:12:6G}$ ACE is bolded. **d** Response of solution FRET ATP DNA DA constructs to ligand. Green box highlights the signaling range of DAs to 10 mM ATP. The response of all DAs was in line with their $k_{off}$ and $k^*_{off,10\,mM}$ values measured by ACE-Scan, with (1), (2), and (3) outperforming the 5′Q$_{1A:15}$ DA (4). The negative control 5′Q$_{-5C:9}$ DA (5) performed poorly, as expected. Error bars represent one standard deviation (N = 3 replicates)

DAs are expected to behave similarly when modeled by a conformational selection-only model. As shown here, in addition to identifying optimal DAs for surface assays, ACE-Scan landscapes can qualitatively direct the design of homogenous solution-phase DAs.

**ACE-Scan of DNA and RNA aptamers with stable structures**. Having validated the performance of ACE-Scan on the classical ATP DNA aptamer, we sought to evaluate whether induced fit binding is found in other DA families. We chose to carry out ACE-Scan on an RNA aptamer that also binds ATP[29], as well as on a DNA aptamer against cocaine[30], both of which form stable secondary structures in the absence of ligand and bind their cognate ligand with a 1:1 stoichiometry. The RNA ATP aptamer adopts a defined, stable secondary structure[29,31] (more thermodynamically stable than the ATP DNA aptamer), and the ligand-bound structure of the ATP RNA aptamer is known, with a G-bulge, base stacking, and non-canonical base pair interactions forming a ligand-binding pocket resembling a GNRA-like motif, in which ATP fills in as the fourth base[31,32] (Fig. 4a). The cocaine DNA aptamer forms a stable planar secondary structure[33,34] with a conserved 3-stem ligand-binding core that binds cocaine (and some cocaine derivatives) with high affinity[30] (Fig. 4b), and is often used to develop proof-of-concept DA biosensors.

The hybridization affinity of ACEs to these aptamers did not increase monotonically with length, as was observed for the ATP DNA aptamer, which can be ascribed to the stable secondary structure of these aptamers interfering with DA formation (Fig. 4c, d). For the ATP RNA aptamer, bases upstream of U16, ahead of the G17-G34 reverse Hoogsteen base pair, act as toeholds to permit ACE duplex nucleation, with experimental hybridization signals correlating with predicted duplex free energies and ACE self-hybridization propensity (Supplementary Fig. 8). However, ACEs with 5′ duplex start sites downstream of C15 hybridized weakly for any given ACE length, visible as a striking boundary in the hybridization heat map (Fig. 4c). For the cocaine DNA aptamer, short (8- to 10-mer) ACEs targeting the P1 and P3 stems, and to a lesser extent the P2 stem (which has a higher GC content that increases duplex stability), were unable to form stable surface-based DAs, consistent with a competition between ACE hybridization and aptamer self-hybridization (Fig. 4d). The hybridization affinity profiles obtained for these DA families illustrate that ACEs nucleate hybridization with aptamers in a highly structure-dependent manner, which is consistent with microarray-based hybridization studies of functional RNAs[17].

ATP RNA DAs displayed relatively low dissociation rates ($k_{off}$) under buffer-only conditions, except for a small group of ACEs hybridizing near G7 ($5′_{7G:8}$ to $5′_{7G:12}$ and nearby ACEs) that showed moderate dissociation (Fig. 4e). Conversely, cocaine DNA DAs displayed high $k_{off}$ rates that were dependent on the location of stems in the native aptamer structure, mirroring the DA hybridization landscape (Fig. 4d, f).

When incubated with high ligand concentrations (10 mM ATP or 100 μM cocaine), ATP RNA DAs exhibited induced fit binding (Fig. 4g) − akin to ATP DNA DAs, whereas cocaine DNA DAs displayed no induced fit (Fig. 4h). For ATP RNA DAs, 7- to 9-mer ACEs hybridizing to the 5′ aptamer extension showed up to a 6-fold increase in duplex dissociation rates as compared to buffer-only conditions, and some 7- to 12-mer ACEs duplexing the GNRA-like binding pocket showed a more than a 10-fold increase. Single-base-mismatched 12-mer ACEs revealed a similar, but higher magnitude, induced fit landscape, a trend that was also observed for ATP DNA DAs (Supplementary Fig. 9). In contrast, single mismatched ACEs for cocaine DNA

DAs did not promote induced fit, instead only increasing $k_{off}$ (Supplementary Fig. 10).

Our observation that cocaine DNA DAs lack an induced fit ligand-binding pathway is consistent with the conformational selection-only binding mechanism originally proposed for cocaine DAs[8]. Although many native aptamers bind their ligand via induced fit, the native cocaine DNA aptamer appears to bind via conformational selection, with the ligand-free and ligand-bound aptamer structures being highly similar[33]. Given this observation, it seems likely that cocaine as a ligand may not be recognized at all by an ACE-duplexed cocaine DNA aptamer, as a DA is expected to share limited structural features with the native aptamer. Furthermore, the absence of induced fit in cocaine DNA DAs may be a consequence of the properties of the native aptamer used to engineer the DA family. Given the limited number of small-molecule-binding aptamers studied here, it will be interesting to explore DAs engineered from other aptamers to help establish criteria that predict conformational selection or induced fit binding in DA families, and to better understand in what manner DA families mirror the binding properties of native aptamers.

**ACE-Scan of a human protein-binding aptamer**. Next, we investigated whether ACE-Scan could be applied to study aptamers against ligands with larger binding surfaces, such as proteins. As a model system, we chose the well-studied thrombin-binding DNA aptamer[35] (TBA) that recognizes human alpha thrombin, a 36.7 kDa protein centrally involved in the coagulation cascade. TBA binds thrombin exosite I with high affinity by assuming a stacked G-quadruplex structure[36], stabilized by a single sodium or, more favorably, potassium ion[37]. TBA was extended with 10 bases at both the 5′ and 3′ ends for this experiment (Fig. 5a). We used thrombin site-specifically conjugated with streptavidin-Alexa Fluor 647 (Methods section), which allowed us to simultaneously monitor the formation of ternary thrombin-DA complexes on the microarray surface during ACE-Scan.

The hybridization affinity of ACEs to extended TBA correlated with predicted hybridization free energies, except for ACEs targeting the 3′ extension, which displayed a lower affinity (Fig. 5b and Supplementary Fig. 11). When subject to sodium buffer-only incubation (Fig. 5c), ACEs complementary to the TBA consensus sequence displayed low $k_{off}$ rates, indicating that the aptamer favored hybridization over internal G-quadruplex formation. Interestingly, ACEs with limited complementarity to the TBA consensus sequence formed DAs with higher dissociation rates under sodium buffer-only incubation. These ACEs are expected to permit TBA G-quadruplex formation, which may lead to steric hindrance or an increase in electrostatic repulsion of TBA from the microarray surface. Such G-quadruplex formation-dependent factors may also be responsible for the weak hybridization affinity observed for ACEs to the 3′ aptamer extension (Fig. 5b).

When incubated with 2 μM thrombin, significant induced fit was observed for DAs with 7- to 9-mer ACEs, as well as longer ACEs with T-5, T3 and G11/T12 5′ start-sites (Fig. 5d). Interestingly, these DAs are expected to have G-quadruplexes that are partially, or completely, duplexed by the ACE. Normalized binding of thrombin to DAs on the microarray surface (Methods section) is shown in Fig. 5e (absolute binding signals shown in Supplementary Fig. 12). Normalization of the AF647 signal accounts for differences in the relative number of DA molecules present on each ACE-specific microarray spot after thrombin incubation, providing a measure of the affinity of non-duplex-disrupted TBA DAs for thrombin. Thrombin binding mirrored the $k^*_{off,2\,\mu M}$ induced fit landscape of TBA DAs, with the

highest binding observed for short ACEs that formed DAs with little switching under 2 μM thrombin incubation, indicating that these DAs retain the ability to bind thrombin while duplexed. We also used potassium, instead of sodium, as a cation in the buffer to investigate the impact of G-quadruplex stabilization on TBA

DA binding. We observed an increase in $k_{off}$ rates and an increased and broader induced fit profile with potassium as a cation (Supplementary Fig. 13), which is consistent with potassium promoting TBA G-quadruplex formation and stabilizing thrombin-TBA binding[37].

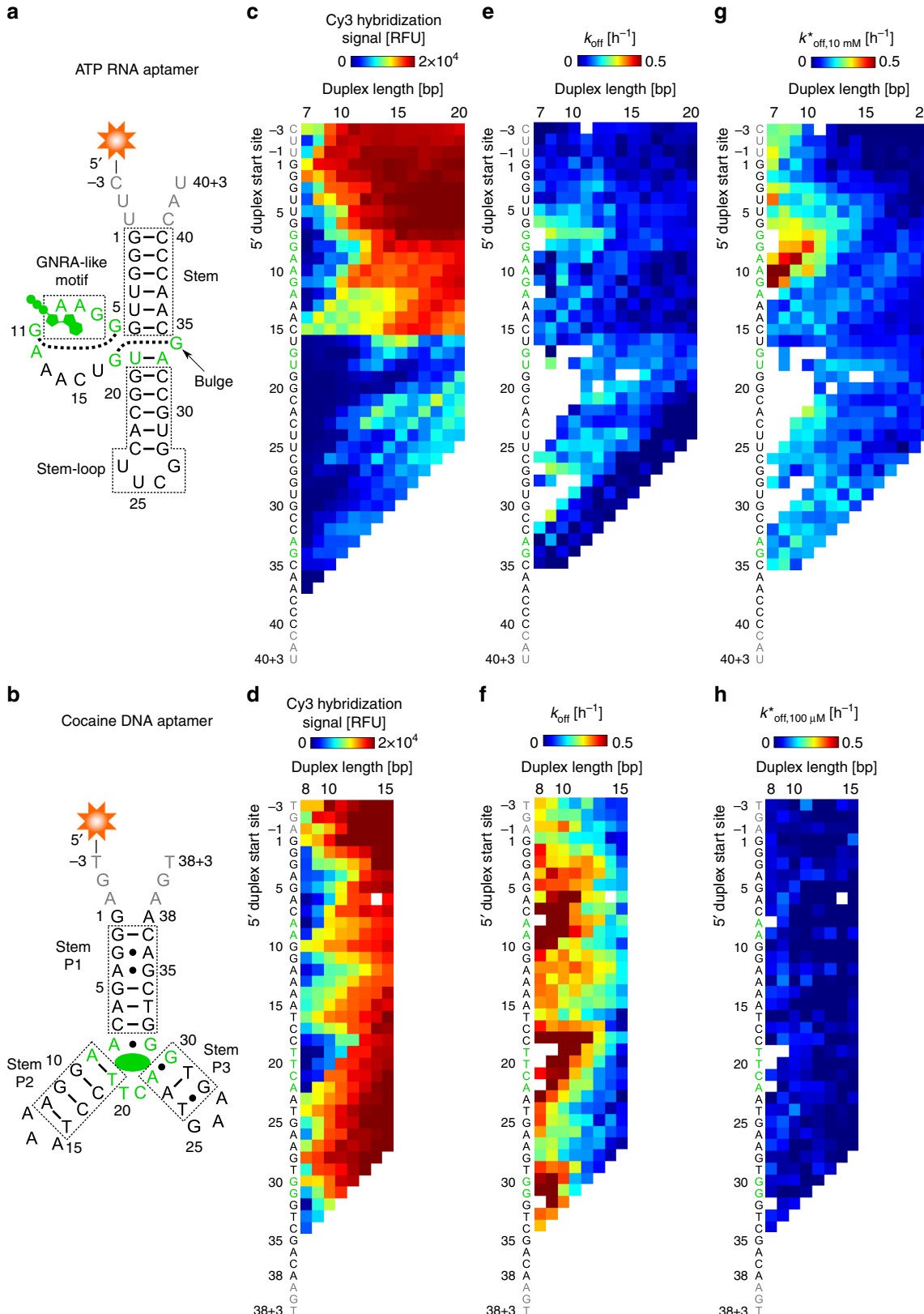

As observed for DAs against small molecules, single mismatched ACEs increased induced fit in TBA DAs (Fig. 5f), with the $k^*_{off,2\,\mu M}$ landscape dependent on the mismatched aptamer base (Fig. 5g). Increased induced fit was observed for ACE mismatches to the TT loops that interface with thrombin, to the G11 and G14 bases involved in proximal G-quadruplex formation, and to the 3′ extension (Fig. 5f, g). It is possible that mismatches to these bases yield DA structures that facilitate initial DA melting or that permit thrombin exosite I to orient more favorably with duplexed TBA, thereby promoting induced fit.

Taken together, our findings suggest that thrombin can reshape and bind TBA DAs in which the G-quadruplex is partially or completely hybridized via an induced fit mechanism. Induced fit may, therefore, be considered the consequence of a chaperone-like[38] activity of thrombin for duplexed TBA, in which initial interfacing of thrombin with TBA DAs leads to re-orientation and remodeling of the contacting surfaces, catalyzing duplex disruption. The positively charged amino acids on the surface of thrombin (e.g., Arg75 and Arg78) may help to initially orient thrombin in proximity to ACE-duplexed TBA, perhaps in coordination with hydrophobic residues (e.g., Tyr76); such coordination between amino acids and nucleic acids has been previously described[39].

**Induced fit in DAs engineered from a natural *add* riboswitch.** After profiling four aptamers discovered through in vitro selection, we applied ACE-Scan to profile a natural aptamer. We selected the *add* riboswitch aptamer from the pathogenic bacteria *Vibrio vulnificus*, which binds adenine and regulates translation of the *add* gene. The *add* riboswitch is a member of the purine riboswitch family[40], which have been the focus of several structural[41] and mechanistic[42–47] studies and that display an interplay between conformational selection and induced fit mechanisms[43,44]. The *add* riboswitch is somewhat unique in that it adopts two conformationally distinct secondary structures in the absence of ligand (apoA, apoB), as well as a stable holo structure induced by the binding ligand, which allows the riboswitch to bind adenine, and regulate translation, in a cation ($Mg^{2+}$) and temperature-dependent manner[46,47]. However, only the apoA state has been documented as capable of binding adenine[46,47]. We synthesized an 87 nucleotide *add* construct taken from the natural mRNA transcript, including downstream bases (up to C96), which are responsible for effecting gene regulation via hybridization to the Shine-Dalgarno sequence (Fig. 6a). We carried out ACE-Scan of *add* DAs at two temperatures (23 °C and 10 °C) and across a range of magnesium concentrations (0–12 mM) to elucidate the role of these parameters on *add* DA ligand binding.

The hybridization affinity of 6- to 12-mer ACEs to the *add* riboswitch aptamer at 23 °C was highly dependent on predicted aptamer secondary structure, as observed for the ATP RNA and cocaine DNA DA families (Fig. 6b). For short ACEs, hybridization was strongest at the single-stranded 5′ end of the aptamer, while for 11- and 12-mer ACEs, strong binding was observed at the P2/L2, J2–3/P3 and P5 regions. The hybridization pattern

obtained at 10 °C was similar, but with stronger signals, as expected (Supplementary Fig. 14).

Upon incubation with 12 mM $Mg^{2+}$ buffer at 23 °C, *add* DAs displayed high $k_{off}$ rates, with ACEs targeting the single-stranded P4/P5 region, as well as ACEs with common 3′ start sites near U47 and hybridizing P2 (3′$_{A45-U48:6-12}$), exhibiting the lowest dissociation rates (Fig. 6c). At lower $Mg^{2+}$ concentrations (4 and 0 mM) or lower temperatures (10 °C), the dissociation rate of DAs was increased but followed similar profiles (Supplementary Fig. 15). Many factors are expected to govern the stability of DAs, and these results might reflect an interplay between the decreased stability of the apoA (free) vs apoB (self-duplexed) states of the *add* riboswitch at lower temperatures and $Mg^{2+}$ concentrations[46], as well as the documented sequence-, temperature- and intramolecular- vs intermolecular-dependent stability RNA: RNA vs DNA:RNA duplexes[48].

When subject to a high concentration of adenine (4 mM), a small subset of *add* DAs exhibited moderate rates of induced fit ligand binding that was highly dependent on ACE location, with ACEs hybridizing the 3′ half of the P2 stem, the J2–3 junction (3′$_{U48-U51:7-11}$), the L3 loop, and to a lesser extent the P3 stem and the 5′ portion of the P4 stem, exhibiting induced fit. (Fig. 5d). The $k^*_{off,4\,mM}$ landscape was temperature and $Mg^{2+}$ dependent, with diminished induced fit observed at lower temperature or $Mg^{2+}$ concentration (Supplementary Fig. 16). As observed in other DA families, the $k^*_{off,4\,mM}$ landscape was not predictable based on DA hybridization affinities or the $k_{off}$ landscape, and when mapped at nucleotide resolution (Methods), induced fit in 9-mer *add* DAs did not correlate to riboswitch sequence conservation scores (Supplementary Fig. 17). Although the lack of widespread induced fit in *add* DAs supports prior evidence of the conformationally selective ApoB-to-ApoA mechanism in native *add* riboswitches, the capability of some *add* DAs with duplexed P2, P3, J2–3 or L3 regions to respond to ligand in a temperature- and cation-dependent manner is surprising. These DAs are highly dissimilar in structure to the ligand-sensing, ApoA state, with P2-duplexed *add* DAs instead resembling the self-hybridized ApoB state reported to not sense ligand[46].

## Discussion

We have introduced ACE-Scan and used it to comprehensively map the hybridization affinities, ACE off-rates ($k_{off}$), and induced fit landscapes ($k^*_{off}$) of DA families engineered from five structurally distinct and well-known aptamers (including a natural riboswitch aptamer). ACE-Scan was shown to be versatile, accommodating a range of assay conditions (e.g., temperature, buffer, one- or two-color readout), ACE designs, aptamers and ligands.

Contrary to common perception, our work revealed rich induced fit landscapes in four out of five DA families investigated, thereby supporting a model of DAs in which some ACE-specific constructs actively sense and bind their cognate ligand from the ACE-duplexed state, and further implying that conformational selection alone cannot adequately model DA ligand-binding dynamics in these cases. Although induced fit is not readily

---

**Fig. 4** ACE-Scan of DAs engineered from DNA and RNA aptamers with stable secondary structures. **a, b** Overview of the functional regions and secondary structure of the (**a**) ATP RNA aptamer, and (**b**) cocaine DNA aptamer. The ligand and bases directly involved in ligand recognition are colored in green. Both the ATP RNA and cocaine DNA consensus aptamer sequences were appended with 3 nucleotides at the 5′ and 3′ ends (gray bases), which are expected to be single stranded and serve as extension toeholds for ACE hybridization. **c–h** 5′ heat maps showing the (**c, d**) experimental hybridization affinity landscapes, (**e, f**) buffer-only $k_{off}$ landscapes, and (**g, h**) $k^*_{off}$ landscapes under high ligand concentrations (10 mM ATP and 100 μM cocaine) for 7- to 20-mer ATP RNA and 8- to 15-mer cocaine DNA DAs, respectively. Whereas a rich induced fit ligand-binding landscape is observed for ATP RNA DAs, cocaine DNA DAs do not exhibit induced fit

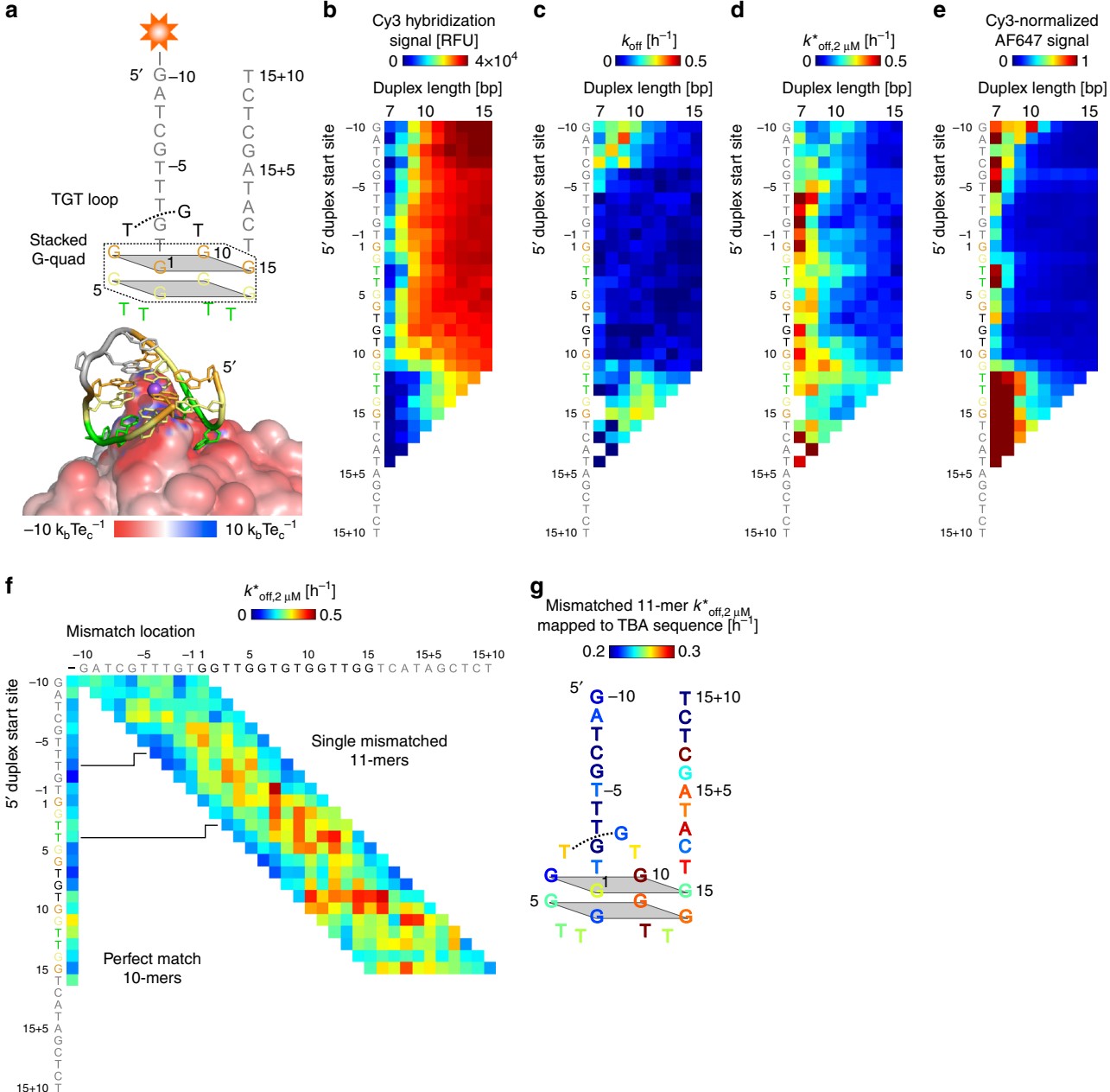

**Fig. 5** Two-color ACE-Scan of TBA DAs. **a** Upper, secondary structure of the TBA construct used here, with 5′ and 3′ extensions colored in gray, TT loops colored in green, and proximal and distal G-quadruplex tetrads colored in yellow and orange, respectively. Lower, 3D view of the interaction of native TBA with human alpha thrombin exosite I (PDB 4DIH[37]). Colored surface shows the APBS-predicted electrostatic potential molecular surface of thrombin, purple sphere represents a sodium ion. **b–d** ACE-Scan generated (**b**) DA hybridization affinity landscape, (**c**) buffer-only $k_{off}$ landscape, and (**d**) thrombin-induced $k^*_{off,2\mu M}$ landscape. **e** Heat map of the Cy3-normalized signal of AF647-labeled thrombin molecules bound to surface-based TBA DAs after incubation with 2 μM thrombin for 1 h. **f** $k^*_{off,2\mu M}$ landscape for DAs engineered with single mismatched 11-mer ACEs. Left-hand column represents perfect match 10-mer ACEs. **g** Linear mapping of the mismatched 11-mer ACE induced fit landscape onto the TBA sequence (Methods section)

predictable a priori, in general, induced fit was maximized by subsets of related 7- to 12-mer ACEs that hybridized to regions of the aptamer containing nucleotides known to interface directly with the ligand. These ACEs may structure DAs such that the aptamer ligand-binding pocket is particularly predisposed for ligand response, thereby allowing the DA to fold through a binding pathway intrinsic to the aptamer, but regulated by the ACE (and other factors, such as buffer cation and temperature). Here, the impact of ligand-induced DA dissociation, as regulated by the ACE (and apparent in the $k^*_{off}$ landscapes), differs from that of spontaneous ACE dissociation, which is regulated by inter

and intrastrand thermodynamics of aptamer and ACE hybridization, and which was obtained by profiling the $k_{off}$ landscape. Interestingly, we note that ACE-Scan may provide a means to study functional nucleic acids, such as aptamers, with ACEs acting as small perturbing elements of the system. Likewise, DAs may be amenable to structural modeling efforts, offering a tractable and uniquely configurable alternative to protein-based systems.

However, perhaps the most straightforward application of ACE-Scan is the development of optimized, high affinity aptasensors engineered from any aptamer of interest. Importantly,

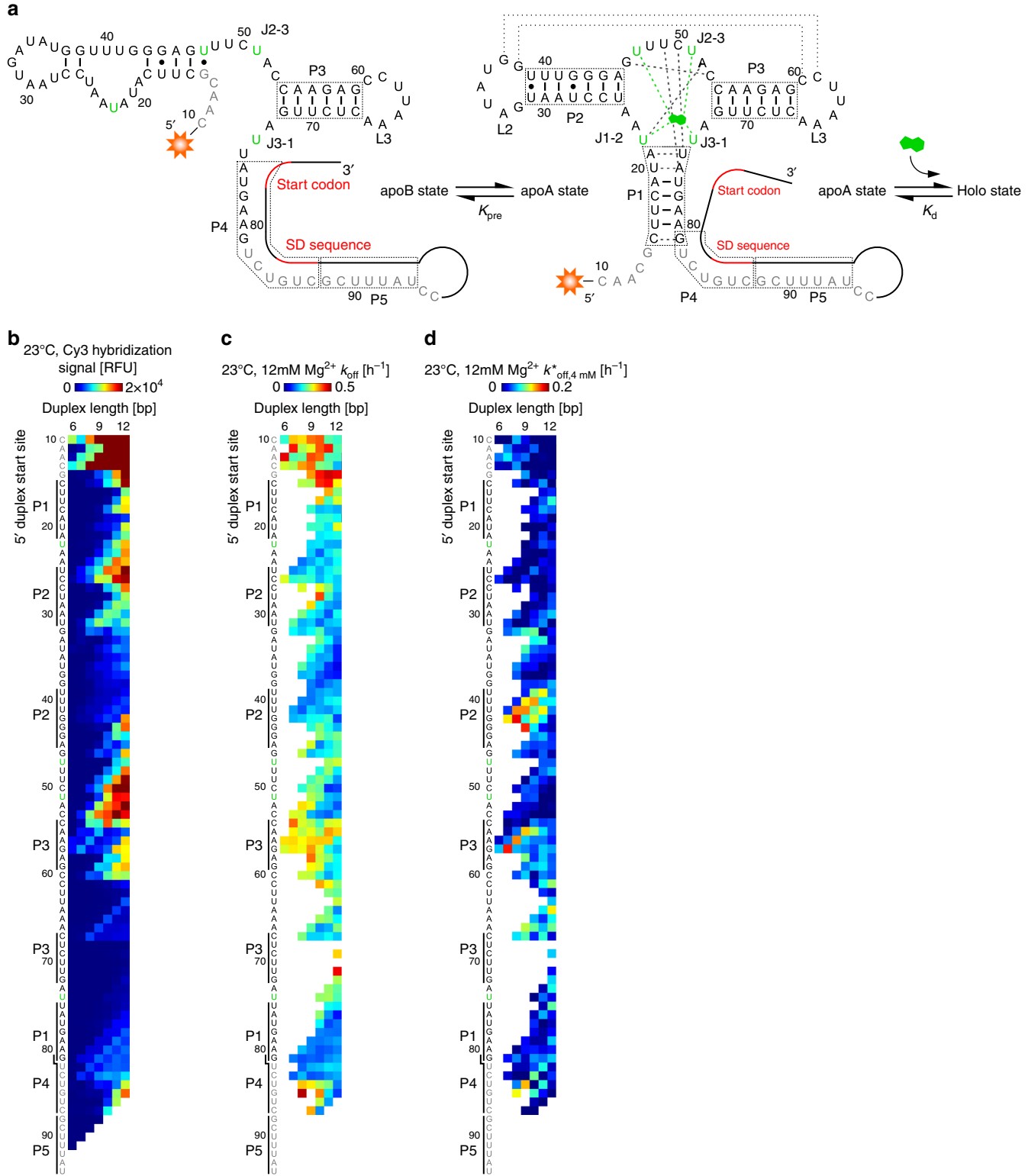

**Fig. 6** ACE-Scan of the adenine-binding *add* riboswitch aptamer. **a** Secondary structure of the apoB (left) and apoA (right) states of the *add* riboswitch aptamer[46]. Stems (P1–P5), loops (L2, L3) and junctions (J1–2, J2–3, and J3–1) are labeled. The location of adenine binding and other functional regions present in the holo structure are shown superimposed on the apoA state as follows: the kissing interaction between L2 and L3 (dotted lines), nucleotides known to interface with adenine (green bases and green dashed lines), and pairs of nucleotides known to interface in the holo state (black dashed lines). Riboswitch elements 3′ of the aptamer involved in translational control were not synthesized here and are drawn schematically as black and red lines. Bases outside the consensus aptamer sequence are colored gray. **b**–**d** Heat maps showing the (**b**) hybridization landscape of add DAs at 23 °C, (**c**) $k_{off}$ landscape of *add* DAs at 23 °C under 12 mM Mg$^{2+}$ buffer-only conditions, and (**d**) $k^*_{off,4\,mM}$ landscape of *add* DAs subject to 4 mM adenine, revealing induced fit binding

each of the nearly 10,000 DAs studied by ACE-Scan here represents a unique surface aptasensor, which can be implemented without modification for fluorescence-based ligand-specific recognition, or adapted to other assay formats and readout modalities. Specifically, to optimize the sensitivity and gain of surface, non-equilibrium DAs, ACEs with high hybridization affinities, low spontaneous off-rates, and maximum induced fit can be readily identified among all possible ACE combinations simply by examining ACE-Scan heat maps. For DAs implemented in homogenous assays (e.g., solution FRET DAs), where both conformational selection and induced fit pathways contribute to signal generation, ACE-Scan can be used to identify ACEs with both high $k_{off}$ and $k^{*}_{off}$ rates, and these ACEs can be directly engineered into DA constructs with improved sensitivity and dynamic range. Here, we show that a handful of ACE-Scan-directed ATP DNA DAs significantly outperform a DA adapted from the literature. Complementing ACE-Scan-directed design, single ACE mismatches can be used as an additional tunable parameter for biosensor optimization. Indeed, although ACEs with single or multiple mismatches have been previously implemented in efforts to improve the biosensing performance of e.g. electrochemical DAs[49], ACE-Scan provides the first, to the best of our knowledge, experimental platform to systematically optimize mismatched DA designs going forward.

ACE-Scan profiles are also relevant to applications in synthetic biology, including the development of improved artificial *cis*-acting riboswitches and *trans*-acting riboregulators. Aptamer-based RNA switches would benefit from the optimization of induced fit, which would allow for artificial switches to react to ligand concentrations closer to the affinity of the non-duplexed aptamer. ACE-Scan may also be effective for identifying regions of an aptamer better suited for the transmission of binding events to effector domains in synthetic constructs. For example, in *add* DAs studied here, although transmission has been evolutionarily conserved to occur via the P1/P4 stem, maximum induced fit was found to be promoted by ACEs hybridizing to the 3′ P2 stem and J2–3 junction that form the aptamer ligand-binding core, a finding that might be leveraged to engineer more sensitive synthetic purine riboswitch constructs. ACE-Scan also complements modeling-based approaches to synthetic biology, such as the thermodynamic-based design of synthetic riboswitches[50], by providing valuable experimental data on DA ligand binding that cannot be predicted beforehand.

Going forward, ACE-Scan, and the principle of applying non-equilibrium microarray-based assays as presented here, are poised for adoption and further modification. For example, ACE-Scan is compatible with NGS flow-cells and live imaging systems, which would allow for the number of ACEs profiled to be further scaled, and also permit real-time DA kinetic measurements, thereby avoiding inaccuracies introduced by the repeated washing and drying of DA microarrays. Our findings also open new opportunities for systematically optimizing ligand-responsive functional nucleic acids that contain hybridizing nucleic acid elements, including aptamer-based biosensors and synthetic riboswitches, as well as DNA nanomachines[51].

## Methods

**Materials and reagents**. All reagents were purchased from Sigma (Oakville, Ontario, Canada), unless otherwise noted. Sodium chloride was purchased from Fischer Scientific (Ottawa, Ontario, Canada). ATP and GTP (100 mM, pre-titrated with NaOH) were purchased from Life Technologies (Burlington, Ontario, Canada). Cocaine (1.0 mg/mL in acetonitrile) was purchased from Sigma (Oakville, Ontario, Canada). Acetonitrile was removed by evaporation, and cocaine was reconstituted in assay buffer. All water used was deionized to 18 MΩ using a Milli-Q system from EMD Millipore (Etobicoke, Ontario, Canada). DNA and RNA aptamers were synthesized by Integrated DNA Technologies (Coralville, Iowa,

USA). All DNA and RNA aptamer sequences, fluorescent modifications, and purification methods are listed in Supplementary Table 1.

Human alpha thrombin bound with a biotinylated active-site inhibitor was purchased from Haematologic Technologies Inc. (catalog # HCT-BFPRCK). Labeled thrombin was prepared by incubating thrombin (2 μM final concentration) with Alexa Fluor 647-conjugated streptavidin (8 μM final concentration, Sigma catalog #21374) in 1xTBA assay buffer for 20 min at room temperature.

Assay buffer for the ATP DNA aptamer consisted of 300 mM NaCl, 5 mM MgCl₂, 20 mM Tris, pH 8.3[3]. Assay buffer for the ATP RNA aptamer consisted of 300 mM NaCl, 5 mM MgCl₂, 20 mM Tris, pH 7.6[29]. Assay buffer for the cocaine DNA aptamer consisted of 140 mM NaCl, 2 mM MgCl₂, 20 mM Tris, pH 7.4[52]. Sodium assay buffer for TBA consisted of 300 mM NaCl, 5 mM MgCl₂, 10 mM phosphate, pH 7.6, whereas potassium assay buffer for TBA consisted of 300 mM KCl, 5 mM MgCl₂, 10 mM Tris, pH 7.6[28]. Assay buffer for the *add* riboswitch aptamer consisted of 130 mM KCl e.g. [45], 0–12 mM MgCl₂, 50 mM Tris, pH 7.5.

**Design of ACE-Scan microarrays**. ACE sequences on the DNA microarray were designed using MATLAB scripts (Mathworks, Natick, Massachusetts, USA, Additional Information). ACE sequences of differing lengths and/or incorporating single mismatches were designed to scan across the entire length of a desired aptamer sequence. The number of possible ACE sequences scales combinatorially; for an aptamer of length $L$, the number of perfect match ACEs of a given length $N$ is:

$$L - N + 1$$

The total number of perfect match ACEs up to a length of $Y$ ($>0$ & $\leq L$) scales with the product of $L$ and $Y$:

$$[Y(2L+1) - Y^2]/2$$

Likewise, the number of perfect match ACEs between lengths $X$ ($>0$ & $\leq Y$) and $Y$ ($>0$ & $\leq L$) scales with the product of $L$ and ($Y–X$):

$$[(Y - X + 1)(2L + 1) - Y^2 + (X - 1)^2]/2$$

The number of possible single mismatched ACEs for ACEs of a given length $Y$ scales with the product of $L$ and $Y$:

$$Y(L - Y + 1)$$

The number of possible single mismatched ACEs for ACEs up to a length of $Y$ scales with the product of ($L–Y$) and $Y^2$:

$$[Y(Y + 1)(3L - 2Y + 2)]/6$$

Likewise, the total number of single mismatched ACEs for ACEs between lengths $X$ ($>0$ & $\leq Y$) and $Y$ ($>0$ & $\leq L$) scales with the product of ($L–Y$) and ($Y^2–X^2$):

$$[Y(Y + 1)(3L - 2Y + 2) - X(X - 1)(3L - 2X + 4)]/6$$

Microarrays were commercially fabricated by MYcroarray (Ann Arbor, Michigan, USA), based on custom 6 × 7k DNA microarrays designs and 3′-to-5′ light-directed synthesis on glass slides with surface densities of $10^{12}$ to $10^9$ reactive amines mm⁻². All ACEs on the microarray were synthesized with an additional 3′ T₂₅ spacer to minimize any influence of the slide surface on aptamer hybridization and DA dissociation[18]. The location of synthesized ACEs on the microarray surface was randomized within a sub-array, and all sub-arrays were synthesized with identical layouts. Depending on the aptamer under study, up to 1400 ACEs varying in ACE length, location, and complementary, with 5–8 replicate probes for each ACE, were synthesized per sub-array. For some ACE-Scan microarrays, ACEs with an additional 5′ T₁₀ linker (T₃₅ total linker length), or 3′ T₁₀-extended ACEs, were included in the microarray design.

In all, 10- to 15-mer ACEs with single mismatches to the aptamer sequence were introduced by using adenosine in place of G, C or T bases in ACEs to create single A–C, A–G, or A–A mismatches with the aptamer, respectively, or by implementing thymidine in place of A bases to form T-T mismatches. Adenosine mismatches were chosen in this work because A–C/G/A mismatches are relatively similar in stability (free energies), minimizing the impact of a particular mismatch identify on DA-binding landscapes[53]. Additionally, the purine ring serves as a strong steric and structural disrupter in adenosine-mismatched DNA duplexes[54], which may promote the dissociation of mismatched DAs.

**ACE-Scan protocol**. To carry out ACE-Scan, first, the fluorescently labeled aptamer of interest was diluted to 0.5 μM in hybridization buffer (4×SSC (600 mM sodium chloride, 60 mM sodium citrate), 0.1% Tween-20, pH 7.0). This solution was heated to 72 °C for 5 min and cooled to room temperature for 15 min. 225 μL of this solution was applied evenly over the entire surface of each microarray, after which microarrays were assembled in custom hybridization chambers, ensuring that no air bubbles were formed during assembly. Microarrays were hybridized in the dark in a humidity-saturated environment at room temperature (except for the temperatures defined for the *add* riboswitch aptamer) for 16 h.

Following hybridization, hybridization chamber assemblies were carefully dissembled while submerged in a 400 mL 2×SSC bath in an ozone-free room. After disassembly, ACE-Scan microarrays were washed in 100 mL 2×SSC for 3 min, and finally washed in 100 mL 1×SSC for 1 min, with all wash buffers at the defined temperature used for hybridization. Microarrays were immediately dried under a stream of nitrogen and imaged in the green (Cy3) channel using an Agilent Technologies G2565CA microarray scanner (Santa Clara, California, USA) in high dynamic range mode with single pass 2-micron resolution, yielding $F_{Hyb}$ for all DAs (Fig. 1c).

After the first imaging round, individual ACE-Scan microarray sub-arrays on a single slide were either (i) left dry (Calibration condition sub-array, Fig. 1c), or (ii) incubated with 80 μl of assay buffer (BufferOnly sub-array, used to calculate $k_{off}$), or (iii) incubated with 80 μL of assay buffer supplemented with the aptamer-specific ligand at the defined temperature used for hybridization (up to 10 mM ATP or GTP for the ATP DNA aptamer, 10 mM ATP for the ATP RNA aptamer, 100 μM cocaine for the cocaine DNA aptamer, 2 μM thrombin for TBA, and 4 mM adenine for the *add* riboswitch aptamer; this sub-array was used to calculate $k^{*}_{off}$ [Ligand]). Incubation was achieved by interfacing the microarray with an Agilent 8-gasket slide, followed by assembling the sandwiched slides in an Agilent hybridization chamber. After 1 h of incubation in the dark at the defined temperature used for hybridization, slides were carefully dissembled while submerged in a 400 mL 2×SSC bath in an ozone-free room, after which the microarray was washed in 100 mL 1×SSC for 1 min, all at the temperature defined for the aptamer under investigation. Following washing, microarrays were immediately dried under a stream of nitrogen and imaged in the green (Cy3) channel, to assay the relative loss of fluorescent aptamers from the surface-based DAs (yielding $F_{Cal}$, $F_{BufferOnly}$ and $F_{[Ligand]}$), and imaged in the red channel (AF647, when using labeled thrombin), to assay the binding of the ligand to non-dissociated surface-based DAs. Images from each fluorescence channel were saved as individual .tiff files.

**Data extraction and analysis.** The fluorescence intensities of individual DA microarray spots within each microarray image were extracted using Array-Pro analysis software (Media Cybernetics Inc., Rockville, Maryland, USA) as the median signal of all pixels within a circular spot boundary. MATLAB scripts were used to compile ACE-Scan data sets into 5′ and 3′ enantio heat map representations (see Data Availability section). Individual DAs were considered of low quality and were removed from the ACE-Scan data set if they had:

(i) low mean hybridization signal, assessed for spot$_i$ as:

$$\text{mean of spot}_i \text{ pixel signals} < 200 \, \text{RFU} \qquad (1)$$

(ii) poor spot morphology (~0.2% of all spots were rejected), assessed for spot$_i$ as:

$$\begin{aligned}
&\text{standard deviation of spot}_i \text{ pixel signals/mean of spot}_i \text{ pixel} \\
&\text{signals} > \text{mean of (standard deviation of pixel signals of all} \\
&\text{spot}_N \text{ in sub-array /mean of pixel signals of all spot}_N \text{ in} \\
&\text{the sub-array)} + 3 \times \text{the standard deviation of (standard} \\
&\text{deviation of pixel signals of all spots}_N \text{ in sub-array /} \\
&\text{mean of pixel signals of all spots}_N \text{ in sub-array)}
\end{aligned} \qquad (2)$$

or (iii) high spot background (~0.2% of all spots were rejected), assessed for spot$_i$ as:

$$\begin{aligned}
&\text{mean signal of pixels surrounding spot}_i > \\
&\text{mean of (signal of pixels surrounding all spots}_N \text{ in sub-array)} \\
&+ 3 \times \text{the standard deviation of} \\
&\text{(signal of pixels surrounding all spots}_N \text{ in the sub-array)}
\end{aligned} \qquad (3)$$

Additionally, any ACE-specific DAs with <3 replicate spots that passed quality control within any sub-array were excluded from the ACE-Scan analysis. For all heat maps shown here, DAs that did not pass quality control are represented as white tiles. The fluorescence intensity reported for DAs that passed quality control was calculated as the mean of the replicate DA spot signals within each microarray sub-array.

For the ATP DNA DA dilution curve data set employing two ACE-Scan microarrays, each microarray contained one Calibration sub-array, one BufferOnly sub-array, and four varying Buffer + Ligand concentration sub-arrays, and the values for Calibration and BufferOnly sub-arrays were averaged across the two microarrays. $K_{Fit}$ and $k^{*}_{off,max}$ values were obtained based on a Michaelis–Menten approximation to the Briggs–Haldane kinetic model of surface DA biosensors[15] by fitting a two-parameter nonlinear regression to the dilution curve of $k^{*}_{off}$ vs. ligand concentration, in the form:

$$k^{*}_{off,[Ligand]} = k^{*}_{off,max}[\text{Ligand}]/(K_{Fit} + [\text{Ligand}]) \qquad (4)$$

Cy3-normalized AF647 signals for TBA DAs were obtained by dividing the red AF647 fluorescence signal for each DA, acquired after incubation with 2 μM

labeled thrombin for 1 h, with the green Cy3 fluorescence signal acquired for the same DA after thrombin incubation.

**Signal calibration procedure.** DA-specific dissociation rates were assessed based on the decrease in surface fluorescence arising from incubation of DAs with varying buffer or buffer + ligand conditions, as normalized to the surface fluorescence signal obtained for a control sub-array that underwent all experimental steps except buffer incubation (Calibration sub-array). Normalization of data sets to the Calibration sub-array accounts for the increased dissociation rate of DAs arising from the repeated drying and washing of the microarray surface during ACE-Scan imaging steps ($F_{Hyb} - F_{Cal}$, Fig. 1c) and corrects for any photo bleaching or experimental variations during microarray handling and imaging.

To normalize data sets, first the overall fluorescence loss between imaging rounds for each DA on each sub-array was calculated by subtracting Calibration, BufferOnly, or Buffer + Ligand sub-array DA fluorescence intensities from the fluorescence intensities measured after hybridization for the same DA. Next, for each DA, BufferOnly and Buffer + Ligand fluorescence losses were divided by the fluorescence loss of the Calibration sub-array. Using this calibration procedure, the relative fluorescence change of the Calibration condition is set to 100% ($F_{Cal} \equiv$ 100%), any additional losses due to buffer incubation are defined as $k_{off}$, and any additional losses due to buffer + ligand incubation are indicative of induced fit binding ($k^{*}_{off}$), as shown graphically in Fig. 1c.

Although we did not correct for the impact of dsDNA structure on Cy3 quenching[55], for the majority of DA constructs studied here, the 5′-located Cy3 label is in a single stranded environment, and therefore the effect of quenching on the hybridization landscapes should be limited. Importantly, all $k_{off}$ and $k^{*}_{off}$ landscapes presented here are based on relative changes in fluorescence for each unique DA, and are therefore not impacted by differences in quenching between constructs.

**Free energy calculations.** DNA and RNA hybridization free energies and ACE self-hybridization free energies were calculated using the DINAMelt webserver[56] using the buffer conditions described for each aptamer and each incubation step. For DNA–RNA DAs, duplexes were modeled as DNA:DNA to enable the effect of differing $Na^+$ and $Mg^{2+}$ concentrations to be taken into account. However, given that most of the DNA:RNA DAs engineered here have moderate purine-to-pyrimidine ratios between DNA and RNA strands, this simplification likely leads to slightly depressed predicted hybridization free energies (and hence lower predicted hybridization affinities) reported here for DNA:RNA DAs[57].

**Solution-phase FRET DA assays.** The Cy3-labeled DNA aptamer and each BHQ2-labeled ACE were reconstituted to 100 μM in water, aliquoted, and stored at −20 °C. The concentration of DNA stocks was determined using a Nanodrop 2000 UV-Vis spectrometer (Nanodrop, Wilmington, Delaware, USA). Each ACE-specific DA was formed in solution at a 3:1 Q:F ratio, using 1.2 μM quenching ACE and 0.4 μM aptamer in ATP DNA DA assay buffer containing 0.1% Tween-20. The DA stock solutions were heated to 72 °C for 5 min, 41 °C for 5 min, and then equilibrated at 22.5 °C for 45 min prior to solution FRET assays to allow DAs to hybridize.

To carry out solution FRET assays of ATP, stock solutions of ATP were prepared by diluting ATP from 20 mM to 0.01 mM in ATP DNA DA assay buffer containing 0.1% Tween-20. 5 μL aliquots of each ATP stock were added into wells of a low-volume black 384-well non-binding microplate (Corning #3676, Corning, New York, USA) on ice, then equilibrated at 22.5 °C for 30 min. The assay was initiated by adding 5 μL of each DA to the ATP aliquots and mixing by pipetting. The microplate was incubated at 22.5 °C for 20 min, and florescence measurements were carried out using a SpectraMax i3x multimode plate reader (Molecular Devices, Sunnyvale, California, USA) in top-read mode, with a 550 nm wavelength fluorescence excitation source (9 nm bandwidth) and 578 nm wavelength fluorescence detector (15 nm bandwidth), maintained at 22.5 °C. The assay contained three replicates of each condition studied, and three fluorescence measurements from the plate reader were recorded and averaged for each replicate. The results were plotted and fit in GraphPad Prism (Graphpad Software Inc., La Jolla, California, USA) using a sigmoidal, four parameter logistic regression representative of log(dose) vs. response.

**Linear mapping of induced fit DA-binding landscapes.** In contrast to 5′ and 3′ enantio heat maps, where individual DA dissociation rates are directly plotted for each ACE, the forward linear mapping of binding landscapes onto an aptamer sequence (or conversely, the reverse linear mapping of individual aptamer base scores onto a 5′ or 3′ enantio heat map representation of DAs) requires an averaging of contributing ACEs (or conversely, aptamer bases) for each base in the aptamer (or ACE) sequence. As such, linear mapping in either direction results in a smoothed data set compared to the single-ACE-based 5′ and 3′ enantio heat map representations shown here.

For forward mapping of 11-mer single mismatched TBA DAs onto the TBA aptamer, each base in the aptamer sequence was colored according to the average induced fit propensity of all 11-mer ACEs that contained a mismatch to that

specific aptamer base. This is equivalent to projecting a weighed average of the columns shown in the $k^*_{off,2\ \mu M}$ heat map onto the aptamer sequence (Fig. 5f).

Sequence conservation scores for the *add* riboswitch aptamer were calculated using Skylign based on the seed sequences available in the Rfam 12.1 database[58] (RF00167 family). 9-mer induced fit landscapes were forward linearly mapped to each base in the aptamer sequence by averaging the induced fit dissociation rate of all 9-mer ACEs with complementary to that base. Likewise, sequence conservation scores could also be reverse linearly mapped to each ACE in a 5′ manner by averaging the information content of each base in the aptamer that was duplexed by each ACE.

**Code availability**. Scripts for creating and analyzing ACE-Scan microarrays, and all experimental datasets, are available on a public GitHub repository: https://github.com/jmunzar/ACE-Scan.

**Data availability**. Microarray datasets are deposited on ArrayExpress under accession numbers E-MTAB-6374, E-MTAB-6375, E-MTAB-6376, E-MTAB-6377, E-MTAB-6378, and E-MTAB-6380. The authors declare that the data supporting the findings of the study are available in the article and the Supplementary Information file, or from the corresponding author upon request.

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

## Acknowledgements

We thank Profs. Robert Sladek (McGill University), Hanadi Sleiman (McGill University) and Alexis Vallée-Bélisle (Université de Montréal) for helpful discussions, and members of our research lab for critical feedback and help with the experimental design and data interpretation. This work was funded by the Canadian Institutes of Health Research (CIHR), the Natural Sciences and Engineering Research Council of Canada (NSERC), the India-Canada Centre for Innovative Multidisciplinary Partnerships to Accelerate Community Transformation and Sustainability (IC-IMPACTS), and the Canada Foundation for Innovation (CFI). J.D.M. was supported by an NSERC Canada Graduate Scholarship, and D.J. acknowledges support from a Canada Research Chair.

## Author contributions

Initial idea: J.D.M., A.N. and D.J. Experimental design: J.D.M., A.N. and D.J. Experimental implementation: J.D.M. and A.N. Data analysis: J.D.M. Project coordination: D.J. Manuscript writing: J.D.M., A.N. and D.J.

## Additional information

**Competing interests:** The authors declare no competing financial interests.

