## [Peer Review File · Nature Communications]

Reviewers' comments:

Reviewer #1 (Remarks to the Author):

The authors demonstrate an array-based assay for evaluating thousands of different duplex constructs for aptamers. Authors achieve this by generating a microarray containing a range of short (7 to ~30 base) oligos that bind to different parts of the aptamer. The aptamer is then fluorescently labeled and hybridized to these short oligos on the array surface. Then the authors measured the dissociation rate between aptamer and the oligo in the presence or absence of the aptamer's ligand. This method was used to generate dissociation-rate maps for 4 different published aptamers and one well-known riboswitch. The central finding of this work is the observation that induced fit (where ligand binding catalyzes duplex dehybridization) appears to occur more frequently than commonly believed (in 3 out of the 4 aptamers and in the riboswitch). The work is original and represents an advancement in the field. However, it suffers from two major issues as described below. A major revision is recommended.

Strengths:

This work is innovative and represents an advancement in the field. The authors have previously described the core method pertaining to surface-based assays of induced fit in duplexed aptamers, but only for a handful of constructs. Here they demonstrate a high-throughput method that effectively profiles the full sequence space of possible complementary sequences. The authors' technique appears to provide considerably more information about the binding kinetics of duplexed aptamers than is available by other techniques, and provides an accessible and high throughput method for profiling the full sequence space of complementary oligos to existing aptamers.

Weaknesses:

While the authors discuss potential applications of their method (in biosensing and in synthetic biology), they do not demonstrate that insights afforded by their method would actually be useful in terms of improving the performance of biosensors or riboswitches. The work would be complete if they could provide an experimental demonstration of an improved biosensor or riboswitch in a "real-world" format (molecular beacon, DNA logic, etc) based on the results of their analysis.

Another major issue is that the data in the paper is presented in a sophisticated fashion but is difficult to interpret, partly due to the authors' failure to include clear descriptions of the underlying analysis and confusing figure labels. For example, authors do not clearly explain the relationship between measured fluorescence (F_{Hyb} , F_{Buffer} , etc), calculated off-rates (k_{off} and k^*_{off}), and the units actually presented in the heatmap figures ("Switching under XXX [%]"). Although the authors explain how to relate fluorescence and off-rates in their first paper (Ref 15), they do not explain it clearly in the main text or methods of this paper, making it impossible for the reader to understand they have calculated the values they present. It is also unclear how k_{off} and k^*_{off} relate to the units in the figures – nowhere is the relationship between off-rate and "% switching" explained, and it is unclear why the range for "% switching" is 0-50. Does "switching" refer to dehybridization only, or also to target binding? Furthermore, the authors do not explain the relationship between "% switching under buffer only" and "% switching under XX ligand concentration". Is each of these calculated relative to the "wash-only" signal, or is the ligand-induced switching calculated relative to the buffer-only signal? The general ambiguity in explaining their experimental procedures makes it difficult to interpret key results of the work (such as the lack of induced fit in the cocaine aptamer shown in Fig 3h). Similarly, Fig 1c, steps 2b and 2c are labeled with "10-30%" and "0-70%". These percentages are not explained in the caption or text and it is impossible for the reader to understand what is meant. The authors should thoroughly revise the manuscript accordingly.

Reviewer #2 (Remarks to the Author):

In this work, Junker and co-workers developed a DNA microarray based method to study the binding mechanism of a few duplex aptamers to test the binding mechanism. With the microarray, a huge number of constructs can be systematically tested and this throughput cannot be accessed previously by traditional assays, or even high throughput methods. I think this method itself is useful and powerful. As to the conclusions obtained, some more fundamental discussion and calculation and comparison with existing literature would make this a stronger paper. In general, I recommend publication of this work, and I have the following comments for the authors to address.

- 1). It is interesting to note that the cocaine aptamer appears to behave differently from the rest, but I'd like to see some efforts, and at least some reasonable discussion on the reason behind this. To me, it is not very obvious based on the current discussion.
- 2). Related to the above point, the authors observed four out of five of the studied aptamers used induced-fitting to bind. However, it is probably still not statistically safe to say that most aptamers use this mechanism. Some thermodynamic discussion to rationalize this observation (as a separate section) would be helpful to convince readers if this is indeed likely to be the case. The use of mismatched sequence is a good data for this purpose as well.
- 3). The authors mentioned for the ATP RNA aptamer and the cocaine DNA aptamer "both of which form stable secondary structures". Are there any citation for this claim, and a brief elaboration on this such as the techniques used to support such conclusions would be very helpful.
- 4). The site 1 and 2 of the ATP DNA aptamer discussion is interesting and its binding thermodynamics was recently studied by ITC and the binding of each individual site was also studied. It might be useful to make some discussion related to this work (DOI: <https://doi.org/10.1093/nar/gkx517>), and see if further insights can be gained.
- 5). In this final part, the authors proposed to use the method for various applications, and I think the optimization of biosensors is a really good idea. Why not demonstrate an improved sensor based on the current data, and I think this will make it a much stronger paper. My feeling is that the fundamental understanding from this paper is still somehow limited. The method appears powerful and potentially useful, but if readers can gain some practical sensors better than the current ones, it is still very attractive.

Reviewer 1 Comments

R1.1: While the authors discuss potential applications of their method (in biosensing and in synthetic biology), they do not demonstrate that insights afforded by their method would actually be useful in terms of improving the performance of biosensors or riboswitches. The work would be complete if they could provide an experimental demonstration of an improved biosensor or riboswitch in a “real-world” format (molecular beacon, DNA logic, etc) based on the results of their analysis.

We agree with the reviewer that the paper would be strengthened by a real world example of an improved biosensor based on ACE-Scan datasets. To this end, we re-engineered five ATP DNA DAs from our ACE-Scan experiments as solution-phase, molecular-beacon-type DA biosensors, tested them in a classical solution-phase FRET assay, and validated that the performance of these DAs is in good agreement with expectations based on ACE-Scan of the ATP DNA aptamer. As shown in the new Figure 3 and accompanying subsection of the manuscript, ACEs predicted to perform well as solution-phase biosensors (based on ACE-Scan) displayed good biosensing performance. Furthermore, the best ACEs outperformed an ACE taken from the literature (with up to 8-fold increased dynamic range), in-line with expectations based on the ACE-Scan dataset.

We would also like to clarify that the main contributions of this paper remain, including (i) describing ACE-Scan, which represents a novel ternary-format assay that is highly relevant to researchers working with high throughput assays or with functional nucleic acids, and (ii) demonstrating the prevalence of induced fit in DAs engineered from highly diverse DNA, RNA and natural aptamers, a finding that contradicts the long-held assumption of conformationally selective binding in the duplexed aptamer biosensing field.

R1.2: Another major issue is that the data in the paper is presented in a sophisticated fashion but is difficult to interpret, partly due to the authors’ failure to include clear descriptions of the underlying analysis and confusing figure labels. For example, authors do not clearly explain the relationship between measured fluorescence (F_{Hyb} , F_{Buffer} , etc), calculated off-rates (k_{off} and k_{off}^*), and the units actually presented in the heatmap figures (“Switching under XXX [%]”). Although the authors explain how to relate fluorescence and off-rates in their first paper (Ref 15), they do not explain it clearly in the main text or methods of this paper, making it impossible for the reader to understand they have calculated the values they present.

Indeed, as pointed out by the reviewer, we had relied on our previous publication to serve as a resource on these points. As requested, we have clarified and improved the descriptions of these topics in the manuscript, which allows the paper to better stand alone. We also note, whereas the relationships between fluorescence and K_{Hyb} , k_{off} , and k_{off}^* were documented in the text and methods sections, they stood to be improved. We have clarified these concerns in the revised manuscript via improved figures (e.g. Fig. 1c and updated and consistent labels on all heat maps), more complete and accurate descriptions in the text, and a clear and consistent lexicon throughout the manuscript.

R1.3: It is also unclear how k_{off} and k_{off}^* relate to the units in the figures - nowhere is the relationship between off-rate and “% switching” explained, and it is unclear why the range for “% switching” is 0-50.

The “% Switching” values represent differences in Relative Fluorescence, as more clearly shown in Fig. 1c in the updated manuscript. However, since these are more accurately represented by the rates k_{off}

and k_{off}^* (and in units of h^{-1}) we have replaced all instances of “% switching” in the text. The exact ranges plotted for k_{off} and k_{off}^* in the heat maps was chosen to be representative of the datasets. For k_{off} , the value would always fall between 0% and 100% (or 0 and 1 h^{-1}), as 0 represents no dissociation, while 100% (or 1 h^{-1}) indicates that all DAs dissociate under buffer in 1 hour. Since k_{off}^* is measured relative to k_{off} (and as more clearly documented in updated Fig. 1c), it would fall between 0% and (100% minus k_{off}) (or between 0 and $1 \text{ h}^{-1} - k_{\text{off}}$), with 0 representing no influence of ligand on DA dissociation (as in the cocaine DA family), while (100% minus k_{off} , or $1 \text{ hr}^{-1} - k_{\text{off}}$) is the maximum dissociation rate of DAs under the presence of ligand that can be measured for a 1 h incubation (i.e. all DAs dissociate after 1 hour).

R1.4: Furthermore, the authors do not explain the relationship between “% switching under buffer only” and “% switching under XX ligand concentration”. Is each of these calculated relative to the “wash-only” signal, or is the ligand-induced switching calculated relative to the buffer-only signal?

As written in the main text, k_{off}^* (originally titled “switching under XX ligand”) is reported relative to k_{off} , and represents the increase in the dissociation rate due to induced fit ligand binding. We have clarified this confusion via an improved Fig. 1c, better lexicon throughout the text, and clarified wording in the Results and Methods sections in the updated manuscript.

R1.5: The general ambiguity in explaining their experimental procedures makes it difficult to interpret key results of the work (such as the lack of induced fit in the cocaine aptamer shown in Fig 3h).

As stated in responses R1.2-R1.4 above, we have expanded and clarified the experimental procedure. We note that we had initially minimized the details of the methodology reported in our previous paper on DAs (Munzar *et al.*, Chem. Sci. 2017) to allow us to focus on the main outcomes of this manuscript (ACE-Scan, heat maps) and our finding of widespread induced fit across DA families.

R1.6: Similarly, Fig 1c, steps 2b and 2c are labeled with “10-30%” and “0-70%”. These percentages are not explained in the caption or text and it is impossible for the reader to understand what is meant.

The range of values shown in Fig. 1c are meant to be generally representative of our ACE-Scan experimental results. This has been clarified and simplified in the revised manuscript (Fig. 1c and accompanying caption text).

Reviewer 2 Comments

R2.1: It is interesting to note that the cocaine aptamer appears to behave differently from the rest, but I'd like to see some efforts, and at least some reasonable discussion on the reason behind this. To me, it is not very obvious based on the current discussion.

We also find this point interesting, and it may largely lie with the mode of action of the native cocaine aptamer. Recent studies have suggested that cocaine intercalates within the folded aptamer structure, and therefore the ligand may not be recognized at all by an ACE-duplexed aptamer, as a DA is expected to share limited structural features with the native aptamer. We briefly touched on this

comment in the original manuscript, and we have now expanded on this aspect. It is interesting to consider that the existence of induced fit within a DA family may be attributable either to the binding structure of the aptamer-ligand complex (being highly structured vs. planar/lock-and-key) and/or to the known binding dynamics of the native aptamer (CS. vs. IF. vs mixed), and these aspects might be fruitful avenues for future studies to pursue.

R2.2: Related to the above point, the authors observed four out of five of the studied aptamers used induced-fitting to bind. However, it is probably still not statistically safe to say that most aptamers use this mechanism. Some thermodynamic discussion to rationalize this observation (as a separate section) would be helpful to convince readers if this is indeed likely to be the case. The use of mismatched sequence is a good data for this purpose as well.

We have softened our discussion to point out that we have only studied a limited sample of the large number of known aptamers. In support of this statement, however, the aptamers we tested were highly dissimilar from structural, functional and ligand standpoints, lending credence to the notion that induced fit in DAs is likely more widespread than thought previously – in reality, no one had previously considered induced fit in DAs, as existing viewpoints were focused on conformational selection in DAs. Furthermore, as our Supplementary Information illustrates, the use of mismatches generally increased induced fit propensity, however induced fit remained a highly ACE-location-specific phenomena, and was not driven by hybridization thermodynamics (see induced fit vs ΔG relationships in the SI figures).

R2.3: The authors mentioned for the ATP RNA aptamer and the cocaine DNA aptamer, both of which form stable secondary structures. Are there any citation for this claim, and a brief elaboration on this such as the techniques used to support such conclusions would be very helpful.

Both of these aptamers are predicted to form highly stable secondary structures in their respective buffers in solution based on their predicted secondary structures. Additionally, the stability of both aptamers has been previously studied experimentally.

For example, this more recent paper studied truncations of the cocaine aptamer that weaken its ligand-free structure:

<http://pubs.acs.org/doi/abs/10.1021/acs.analchem.6b01633?journalCode=ancham>

And this earlier paper uses NMR and calorimetry to study the stable structure of the consensus cocaine DNA aptamer:

<http://pubs.acs.org/doi/abs/10.1021/bi100952k>

For the ATP RNA aptamer, the original paper clearly shows the highly ordered secondary structure using in-line probing:

<https://www.nature.com/nature/journal/v364/n6437/abs/364550a0.html>

The stable structure of the ATP RNA aptamer was also confirmed in the follow-up NMR work, which found that the loop and bulge regions of the aptamer were the only accessible regions, and only in the absence of ligand:

<https://www.nature.com/nature/journal/v382/n6587/abs/382183a0.html>

Although some of these citations were included in the original manuscript, we have added these four references to the introduction of the ATP RNA and cocaine DNA DA subsection of the Results.

R2.4: The site 1 and 2 of the ATP DNA aptamer discussion is interesting and its binding thermodynamics was recently studied by ITC and the binding of each individual site was also studied. It might be useful to make some discussion related to this work (DOI: <https://doi.org/10.1093/nar/gkx517>), and see if further insights can be gained.

Indeed we are aware of the above paper, and it is interesting to consider the order of binding of ATP molecules to the aptamer and to the DA, and also the implications of engineering DAs with controlled cooperativity/ higher cooperativities. We have added this aspect to the discussion of the ATP DNA DA subsection of the Results.

Perhaps of interest to the reviewer, we would like to point to the enthalpy finding and associated discussion in the above-cited paper:

“It appears adenosine's binding to Apt1b has fewer base pair formation compared to its binding to Apt1a. This could be explained by the fact that Apt1a has only six base pairs, while the rest of the sequences do not form a stable structure in the absence of adenosine. After binding adenosine, a large structural change to a more rigid one is observed. For Apt1b, however, two stable base paired regions exist and a smaller structural change is needed for binding.”

In this regard, we note that, following the authors' own logic in the above paper (and using their notation for Binding Sites in the native ATP DNA aptamer), Site 2 would be expected to close before Site 1 in the native aptamer, and that while each site binds with the same affinity, this might give a clue as to the order of cooperative binding in the native aptamer. This idea is not expanded upon in the publication - however, and interestingly, we note that ATP DNA DAs exhibit significant induced fit binding for ACEs that partially or completely block Site 1 and 2, but only for DAs in which ACEs bind the 5' half of the aptamer (and not for ACEs that leave Site 1 free), and this drops off rapidly once the stem is hybridized and is not reflected for ACEs hybridizing at the 3' end (see the k_{off}^* heat for the ATP DNA DA family, Fig. 2d). It might be that blocking the stem region inhibits the initial formation/binding of ATP to Site 2, and this event must occur before Site 1 binding (and concomitant ACE dehybridization) can take place. While speculative and outside the scope of our current manuscript, this proposed mechanistic pathway for DA-ATP binding is in line with our previously published dataset and discussion (Munzar *et al.*, Chem. Sci. 2017).

R2.5: In this final part, the authors proposed to use the method for various applications, and I think the optimization of biosensors is a really good idea. Why not demonstrate an improved sensor based on the current data, and I think this will make it a much stronger paper.

We agree. Please see our response to Reviewer 1 in discussion point **R1.1** above.

R2.6: My feeling is that the fundamental understanding from this paper is still somehow limited. The method appears powerful and potentially useful, but if readers can gain some practical sensors better than the current ones, it is still very attractive.

We have reiterated the fundamental importance/outcome of our work in the second half of discussion point **R1.1** above. Additionally, given the large improvement in the performance of an ATP DNA DA biosensor that we demonstrate in this updated manuscript, we hope that the importance of induced fit binding in DAs, together with the validation of ACE-Scan-directed DA biosensor design, will allow this manuscript to resonate with a wider audience.

REVIEWERS' COMMENTS:

Reviewer #1 (Remarks to the Author):

In the revised MS, the authors have satisfactorily responded to my prior comments. First, they have improved the clarity and consistency of figure labels so that the results of the work can be understood without referencing the authors' prior work. Importantly, the authors have successfully demonstrated five DAs (based on their ACE scan data for the ATP DNA aptamer) in a molecular beacon biosensor format, and showed that ACE's identified through their method perform as expected in an alternate assay format in solution. Given these improvements, I suggest acceptance of the manuscript without modifications.

Reviewer #2 (Remarks to the Author):

The authors have addressed my comments and I recommend publication.